# Combinational Variation Temperature and Soil Water Response of Stomata and Biomass Production in Maize, Millet, Sorghum and Rice

**DOI:** 10.3390/plants11081039

**Published:** 2022-04-11

**Authors:** Phanthasin Khanthavong, Shin Yabuta, Al Imran Malik, Md Amzad Hossain, Isao Akagi, Jun-Ichi Sakagami

**Affiliations:** 1The United Graduate School of Agricultural Sciences, Kagoshima University, Kagoshima 890-0056, Japan; khanthavongp@gmail.com (P.K.); amzad@agr.u-ryukyu.ac.jp (M.A.H.); 2National Agriculture and Forestry Research Institute, Dong Dok, Ban Nongviengkham, Vientiane 7170, Laos; 3Faculty of Agriculture, Kagoshima University, Kagoshima 890-0056, Japan; syabuta@agri.kagoshima-u.ac.jp (S.Y.); akagi@agri.kagoshima-u.ac.jp (I.A.); 4Alliance of Bioversity International and CIAT (Asia), Lao PDR Office, Dong Dok, Ban Nongviengkham, Vientiane 7170, Laos; a.malik@cgiar.org; 5Faculty of Agriculture, University of the Ryukyu, Okinawa 903-0213, Japan

**Keywords:** *gs*, leaf area, temperature-dependent, transpiration, water stress, water use efficiency

## Abstract

Environmental responses of stomatal conductance (*gs*) as basic information for a photosynthesis-transpiration-coupled model have been increasing under global warming. This study identified the impact of *gs* behavior under different soil water statuses and temperatures in rice, maize, millet, and sorghum. The experiments consisted of various soil moisture statuses from flooding to drying and combination of soil moisture status and temperature. There was a reduction in shoot biomass of maize and sorghum caused by decreasing of *gs*, photosynthesis (*A*), and transpiration (*E*) in early imposed waterlogging without dependent temperature, whereas millet and rice were dependent on temperature variation. The effect of gradual soil drying, *gs*, *A*, and *E* of maize, millet, and sorghum were caused by low temperature, except rice. The impact of the combination of various soil water statuses and temperatures on *gs* is important for the trade-off between *A* and *E*, and consequently shoot biomass. However, we discovered that an ability to sustain *gs* is essential for photo assimilation and maintaining leaf temperature through evapotranspiration for biomass production, a mechanism of crop avoidance in variable soil water status and temperature.

## 1. Introduction

Global climate change increases variability in temperature, drought, and flooding [1,2,3]. Rice (*Oryza Sativa* L.) as C_3_, and maize (*Zea mays.* L.), millet (*Panicum miliaceum* L.), and sorghum (*Sorghum bicolor* (L.) Moench) as C_4_ are cereal crops grown under variable climates and rainfed environments in Asia, America, and Africa, sharing a high contribution to global food security [4]. Climate change and weather disasters are major causes of reductions in agricultural productivity [5,6].

Under rainfed conditions in tropical and subtropical regions, these crops experience diverse individual and successive combined environmental stresses attributed to climate change such as drought, flooding, and temperature variability that directly affect their morphology and physiology, leading to crop failure. The effect of environmental stress such as water stress and temperature on crop production is well documented [7,8,9,10]. This study focusses on to access crops response to combined soil water status and temperature in relation to morphological and gas exchange parameters (i.e., photosynthesis and stomatal behavior).

Stomata are the gatekeepers of gas exchange and the primary determinants of CO_2_ assimilation. Stomata conductance (*gs*) response to soil water and temperature stresses is basic information for photosynthesis transpiration, and it has increasingly been a concern under global warming. The positive correlation between *gs* and photosynthesis (*A*) has been reported in the laboratory [11] and a positive correlation between *gs* and yield has also been reported in field conditions [12]. Alternatively, stomatal closure is caused by water stress and temperature [13]. Additionally, stomatal closure can directly influence CO_2_ absorption (photosynthesis rate) and transpiration rate (*E*) [11,14].

Rice, a C_3_, an original lowland crop, is resilient, and due to its crucial root anatomy to cope with soil waterlogging [15,16] has been introduced to waterlogging and upland conditions [17]. In contrast, maize, millet, and sorghum are better adapted to upland conditions due to their water absorption ability that is related to their deep root system [18]. Nevertheless, the response of crops to soil water status depends on crop genotypes and varieties [19,20].

More than one-third of the world’s irrigated area suffers due to waterlogging. Continuous flood conditions lead to lack oxygen in the soil, restricting respiration of growing roots, living organisms, and changing soil chemical property [21]. The response of crops to waterlogging depends on varieties. Most of the upland crop species are sensitive to waterlogging conditions compared to wetland crop species such as rice due to their inability transport oxygen from the leaves to root tips for sustaining the root respiration and gas exchange. In condition, waterlogged soil cause reduction of *gs* and *A* in sorghum [22,23], maize [24], and millet [9]. Therefore, maize, millet and sorghum reduce in growth and grain yield under waterlogging [9,24,25,26].

Drought occurs when the soil moisture is continuously low, where water extraction by root and water transport within the plant is reduced. To overcome drought stress, plants respond by increasing the water extraction efficiency and the water use efficiency of the root, and simultaneously reduce *E* (water loss) [27] by closing stomata as well as maintaining turgor [28]. Crops have different water requirement for growth development and productivity. Rice and maize had higher water requirement than millet and sorghum [29]. The ability to maintain photosynthesis during drought is indicative of the potential to sustain productivity under water deficit. The stomatal response to drought conditions depends on crop genotypes [30,31]. Sorghum exhibits the ability to maintain stomatal opening and photosynthesis at low water potentials, as well as the ability for osmotic adjustment [32]. In rice, the photosynthetic rate declines dramatically during soil drought, mainly due to the decrease in the *gs* [33]. Stomatal limitation on photosynthesis depends on the level of drought [28].

Extreme temperature directly impacts on the production of cereal crops. The optimal temperature of C_3_ plants (28–30 °C) is lower than C_4_ plants (26–35 °C) such as maize, millet, and sorghum [34,35]. C_3_ and C_4_ plant species show various responses to *gs*, *A*, and *E* under temperature stress [36,37,38]. Increase in global temperature can directly affect stomatal behavior and reduce yield in major crops [7,39]. The increase in mean global temperature has been predicted under climate change [40]. Increasing of temperature is closely associated with increased vapor pressure deficit (VPD). The key response of crops to variation of VPD is by regulation of *E* through *gs* [41]. On the other hand, low temperature is another influence on stomatal aperture of crops. Cool conditions affect stomatal closure in *Phaseolus vulgaris* and maize [42]. Low temperature causes a reduction in the plant’s capacity for calcium uptake by guard cells due to stomatal closure [43,44]. Calcium acts as an intracellular secondary messenger, regulating ion transport activity plasma and vacuolar membranes in guard cell turgor [44,45].

Previously, our study showed that rice and millet have better root distribution under waterlogging than in dry conditions compared with maize and sorghum, whose root distribution was limited under waterlogging, leading to poor growth of aboveground biomass [19]. However, this study was conducted in a specific environment only. A combination of factors such as the variable temperatures, drought, and waterlogging occur during crop production, especially under rainfed agriculture. The effect of combination of factors on crop failure may be higher than an individual factor.

Many studies have reported the effect of combinations of water stress and temperature variability on the growth and productivity of crops [46,47,48,49,50]. However, knowledge on the effect of various soil water status and temperature variabilities such as soil waterlogging, dry conditions, and their interactions with low and high temperatures on stomatal response among crop genotypes are scant. Hence, we hypothesized that the response of shoot biomass and *gs* behavior of different crop genotypes to combinations of soil water stress and temperatures have an effect on crop genotypes due to their variable adaptability of *gs*. Therefore, we identified the variation in stomatal traits and the impact of *gs* behavior under various soil water status and temperatures on rice, maize, millet, and sorghum.

## 2. Results

### 2.1. Experiment 1

#### 2.1.1. Soil Control and Atmospheric Environment

The change in soil moisture content, air temperature, relative humidity, and vapor pressure deficit (VPD) during the treatment period are summarized in Figure 1. The trend of soil moisture for each treatment in experiment. 1A and 1B (Exp. 1A and 1B) was similar, where waterlogging (WL) and dry soil (DH) had the highest (38.8% and 43.7% for Exp. 1A and 1B, respectively) and lowest moisture contents (7.6% and 10.6% for Exp. 1A and 1B, respectively), respectively (Figure 1a,d). Soil moisture content with severe dry soil treatment (DH) gradually declined from 16.0% down to 7.6% for Exp. 1A and 15.1% down to 10.6% for Exp. 1B during the treatment period. The temperature in Exp. 1A was higher than in Exp. 1B with the average temperature of day/night being 34/25 °C and 24/15 °C in Exp. 1A and 1B, respectively (Figure 1b). The vapor pressure deficit, relative humidity, and solar irradiance were not significantly different between Exp. 1A and 1B, but their fluctuations were different between Exp. 1A and 1B (Figure 1c,e,f).

#### 2.1.2. The Correlation between Soil Moisture Status and Shoot Biomass, LA, and *gs*

A linear and nonlinear correlation that depended on crop and experiment existed between soil moisture status and shoot biomass, LA, and *gs* in comparison between Exp. 1A and 1B (Figure 2). There were significant nonlinear correlations between soil moisture status and shoot biomass, LA, and *gs* for both Exp. 1A and 1B (Figure 2a,e,i) in maize. Furthermore, the correlation between soil moisture status and shoot biomass, LA, and *gs* was observed as a nonlinear correlation on shoot biomass, LA, and *gs* in Exp. 1A in sorghum, whereas, in Exp. 1B, it was a linear correlation on shoot biomass, LA, and *gs* (Figure 2b,f,j). In millet, the nonlinear and linear correlation between soil moisture status and shoot biomass, LA, and *gs* was observed in Exp. 1A and 1B, respectively. The correlation was significant between soil moisture status and shoot biomass for both Exp. 1A and 1B, LA for Exp. 1A, and *gs* for Exp. 1A, but Exp. 1B showed no significant correlation between soil moisture status and LA (Figure 2c,g,k). Additionally, a nonlinear correlation between soil moisture status and shoot biomass LA, and *gs* was found in Exp. 1A and 1B in rice, whereas the excluded correlation between soil moisture status and shoot biomass in Exp. 1A showed a negative linear correlation. A significant correlation was found between soil moisture status and shoot biomass, LA, and *gs* for Exp. 1A and 1B (Figure 2d,h,l). The distance of correlation lines between soil moisture status and shoot biomass, LA, and *gs* showed that maize and rice had fewer distance correlation lines between Exp. 1A and 1B than sorghum and millet (Figure 2).

### 2.2. Experiment 2

#### 2.2.1. Crop Response to a Combination of Soil Moisture Status and Temperature on Shoot Biomass, LA, and Gas Exchange

The volumetric soil moisture content of moderate soil moisture (MSM), gradual soil drying (GSD), and continuous soil waterlogging (CSW) combined with low or high temperature is shown in Figure 3. There was less difference between the soil moisture status combinations with low or high temperatures. It is because the soil moisture content was controlled at field capacity before the start of the treatment. After treatment, the average soil moisture content under MSM/24/15 °C or 34/25 °C was maintained at field capacity. In contrast, the soil moisture content under GSD/24/15 °C or 34/25 °C was gradually reduced by withholding irrigation for 17 days. Alternatively, when the pots were submerged, the volumetric soil moisture content under CSW/24/15 °C or 34/25 °C depicted very little change.

The effect of the combination between soil moisture status and the temperature varied significantly depending on crop (*p* < 0.001) for shoot biomass, LA, *A*, *gs, E,* and water use efficiency (*WUE*) (Table 1). Similarly, treatments on shoot biomass showed significant effects, *A*, *gs*, *E*, and *WUE* (*p* < 0.001) for all crops. In contrast, there was no significant effect within the crops on all parameters.

All crops showed a negative response on shoot biomass and LA under MSM with low temperature, except LA of maize showed a positive response under this condition. Under GSD, maize and sorghum had better shoot biomass and LA growth under GSD/34/25 °C compared to millet and rice; maize and rice showed positive response on shoot biomass under GSD/24/15 °C, but not sorghum and millet. Under GSD/24/15 °C, the LA of all crops had a negative response. Moreover, each crop showed a similar response on shoot biomass and LA under CSW/24/15 °C and 34/25 °C. Maize and sorghum had an adverse reaction to CSW, whether 24/15 °C or 34/25 °C. Alternatively, the effect of CSW on the shoot biomass and LA of millet and rice were negatively affected by low temperature (24/15 °C).

Under various combined factors, as presented in Figure 4, there were variations in gas exchange among the crops. The *A* was a positive response in all crops grown under MSM/24/15 °C or 34/25 °C (Figure 4c). Under GSD/24/15 °C, a negative impact existed in maize and millet, and that on rice was under GSD/34/25 °C. In comparison, the negative effect of GSD on *A* of sorghum was found at low and high temperatures. Under CSW, maize and sorghum had a negative response on *A* at low and high temperatures, whereas the effect of CSW in millet and rice was found in low temperatures (Figure 4c). Low temperature harmed *E* of all crops grown under different soil moisture status (Figure 4d). Under MSM, *gs* of all crops had a positive response in high temperatures, but they showed a negative impact at low temperatures except maize. There was a high negative impact on *gs* in maize, sorghum, and millet under GSD/24/15 °C. Furthermore, GSD showed a negative response on *gs* at low and high temperatures in rice. Under CSW, *gs* of maize, sorghum, millet, and rice demonstrated a similar response of *A* with maize and sorghum, harming *gs* at low and high temperatures (Figure 4e). Figure 4f shows that low temperature promoted a positive response of *WUE* under numerous soil moisture statuses, but high temperature negatively impacted *WUE* in all crops.

#### 2.2.2. Changing of Gas Exchange

The effect of the combination between soil moisture status and temperature treatments on gas exchange is shown in Figure 5. There was a significant effect of combination treatments on *A*, *gs*, *E*, and *WUE* at 4, 8, 12, and 17 days after treatment (DAT) (*p* < 0.05) in maize, except for *gs* at 4 DAT that showed no significant difference among the treatments. With low temperature, each soil moisture status had lower *A*, *gs*, and *E* of maize than high temperature. *A*, *gs*, and *E* of maize significantly decreased in low temperature at 4, 8, and 12 DAT under GSD, but declined *A*, *gs*, and *E* were delayed at 17 DAT (Figure 5a,e,i,m) compared to MSM with high temperature (MSM/34/25 °C). Under CSW, *A*, *gs*, and *E* of maize significantly decreased at initial (4 DAT) after imposed soil waterlogging in low and high temperatures compared to MSM/34/25 °C and GSD/34/25 °C; *WUE* of the maize was influenced by low temperature combined with all soil water statuses, particularly MSM and GSD, compared to high temperature. Additionally, a significant effect of treatments on gas exchange was found at 4, 8, 12, and 17 DAT in sorghum. Under different combinations of various soil water status and temperature, the change in gas exchange compared to sorghum and maize was similar. The combination of soil water status and low temperature decreased *A*, *gs*, and *E*, but it increased *WUE*. After the low-temperature imposition, *A*, *gs*, and *E* of sorghum under different soil moisture levels decreased at the inceptive stage. Its *A*, *gs*, and *E* under MSM and GSD recovered at 8 DAT, but not under CSW. The *A*, *gs*, and *E* of sorghum at low and high temperatures gradually declined along with soil moisture status (Figure 5b,f,j,m) under GSD. Nevertheless, the *A*, *gs*, and *E* under GSD combined with low temperature (GSD/24/15 °C) was lower than high temperature (GSD/34/25 °C), and GSD/24/15 °C was not significantly different compared to CSW combined with low and high temperatures. Under CSW, the *A*, *gs*, and *E* was significantly decreased at 4 DAT by CSW, and CSW/24/15 °C and CSW/34/25 °C on *A*, *gs*, and *E* (Figure 5b,f,j,n) showed no significant difference.

There were significant effects of the combination treatments on the change of gas exchange (Figure 5c,g,k,o) in millet. MSM and GSD combined with low temperature were initially lower at *A*, *gs*, and *E* than high temperature at 4, 8, and 12 DAT, but CSW/24/15 °C did not decrease *gs* of millet at 4 DAT compared to the treatment before. Under MSM/34/25 °C, the gas exchange did not change the *A*, *gs*, and *E* at all measured times, but the gas exchange was reduced under MSM/24/15 °C, specifically on *E*. There was a similar reduction of gas exchange of millet under GSD/24/15 °C with MSM/34/25 °C at 4 DAT. In contrast, *A*, *gs*, and *E* under GSD/34/25 °C was delayed to record a significant decrease at 17 DAT. The impact of CSW in *A*, *gs*, and *E* depended on temperature. CSW/34/25 °C showed no significant difference on *A*, *gs*, and *E* of millet compared to MSM/34/25 °C, whereas under CSW/24/15 °C, *A* and *E* was reduced for maize at 4 DAT and *gs* at 8 DAT. Millet responded similar to maize and sorghum, where *WUE* increased under all SMC combined with low temperature (Figure 5o). The effect of treatments on *gs*, *E*, and *WUE* was noticed at all measured times in rice, but *A* was seen at 12 and 17 DAT (Figure 5d,h,l,p). Rice showed the highest *gs* and *E* at 4 DAT under CSW/34/25 °C, but not for *A*. The highest *A* was seen under MSM/24/15 °C from 8 DAT. However, CSW/24/15 °C had lower *A*, *gs*, and *E* than CSW/34/25 °C, and CSW/34/25 °C had greater *gs* from 4 DAT than other treatments. Under GSD combined with low and high temperatures, *A* was shown to be alternative to *gs* and *E*, where it was higher under GSD/24/15 °C than under GSD/34/25 °C at 17 DAT. Although under GSD/34/25 °C, *gs* and *E* was higher at 4, 8, and 12 DAT than GSD/24/15 °C, at 17 DAT, there was no significant difference in *gs* and *E* between GSD/24/15 °C and GSD/34/25 °C.

#### 2.2.3. Correlation between *gs*, *A*, and *E* and Influence of Atmospheric Environment and *gs* on Shoot Biomass

Figure 6 presents the correlation between *gs* and shoot biomass, *gs* and *A*, and *gs* and *E* across the combination between soil moisture status and temperature treatments, which was positively significant for all crops (Figure 6). Maize had the highest coefficient, followed by millet, sorghum, and rice in a correlation coefficient between *gs* and shoot biomass (Figure 6a–d). The correlation between *gs* and *A* in maize had the highest coefficient, followed by sorghum, millet, and rice (Figure 6e–h). In contrast, a high correlation coefficient between *gs* and *E* was found in maize, followed by sorghum and rice (Figure 6i–l).

Multiple linear regression analysis was used to identify which environmental factors and physiological traits influenced shoot biomass across a combination of various soil water statuses and temperatures, *gs*, *A*, and *WUE*. Our results showed that soil moisture content, temperature, and *gs* were suitable parameters to generate a formula that highly contributes to multiple crops. Soil moisture content and temperature influenced *gs* of all crops. The result of multiple linear correlation showed that sorghum had the highest adjustment (Adj) of R squared (Adj. R^2^ = 0.759, *p* < 0.001), followed by maize (Adj. R^2^ = 0.658, *p* < 0.001), millet (Adj. R^2^ = 0.492, *p* = 0.006), and rice (Adj. R^2^ = 0.262, *p* < 0.066) (Table 2). On the basis of β-value, rice and maize were less affected by temperature and soil moisture content compared to sorghum and millet; temperature especially had a higher influence on shoot biomass of sorghum and millet than maize and rice (Table 2).

## 3. Discussion

### 3.1. The gs Responses to Soil Moisture Status and Environmental Influence on Biomass Production

Our study highlighted the interaction between crop genotypes and combination of soil moisture status and environment through *gs* and shoot biomass. Stomatal aperture is influenced by a number of environmental factors including water variability, leaf temperature, and CO_2_. The dynamic of stomatal movement acting in response to environmental charge and internals in an attempt is to optimize the trade-off between *A* and to maintain plant water status (transpiration rate) [51]. Close positive correlation among *gs*, *A*, and plant growth have been found under the control environments and field experiments [11,12,52]. Plant mechanism of response to water stress includes conservative, where the plants close the stomata are faster, and non-conservative, where the plants close the stomata are slower under drought conditions [53]. Our study emphasized on non-conservative mechanism.

The correlation between soil moisture status and shoot biomass and *gs* was a similar tendency (Figure 2). Under wet soil conditions, the shoot biomass of maize and sorghum declined (Exp. 1A and 1B). Additionally, the *gs* of these two crops were limited by wet soil conditions, especially waterlogging in Exp. 1A and 1B (Figure 2). A similar response of shoot biomass and *gs* of maize and sorghum was noticed under waterlogging interaction with low and high temperatures. It showed that maize and sorghum were sensitive to soil waterlogging and were temperature-independent (Figure 6). This finding is confirmed previous reports [23,54,55]. Waterlogging extremely limited root length density at the deep soil layer and shoot biomass of maize and sorghum [19,56] due to their roots suffering from low oxygen diffusion in the soil [57,58,59].

Moreover, shoot biomass of millet showed a negative response to waterlogging in both Exp. 1A and 1B. Still, its impact on shoot biomass under combination of CSW and high temperature was the opposite in experiment 2. The temperature was similar to CSW interaction with a high temperature in experiment 2 (Figure 4a). Barnyard millet adapted well to waterlogging [19,20]; not withstanding, low temperature caused a reduction of shoot biomass under waterlogging in Exp. 1B and low temperature (CSW/24/15 °C) in experiment 2 (Figure 4a). Under the screen house, the fluctuation of light intensity influenced *gs*, *A,* and biomass production [60]. It was reported that under optimum temperature, rice is well adapted to waterlogging [17]. However, sub-optimum temperature (<20 °C) affected reduction of shoot biomass and relative growth in rice compared to optimum temperature [61]. Similarly, the combination of waterlogging and low temperature caused a reduction of shoot biomass and *gs* of rice compared to a higher temperature (Figure 2 and Figure 4a). The shoot biomass and *gs* crop response to dry soil conditions or combination of gradual soil drying and low or high temperature were computed among crops and within the treatment in experiment 1 and 2 (Figure 2 and Figure 4). The correlation trend between soil moisture status and shoot biomass and *gs* in Exp. 1A and 1B or response of shoot biomass and *gs* under combination of gradual soil drying and high or low temperature (Experiment 2) of each crop were similar (Figure 2 and Figure 4a,e). These results imply that *gs* were influenced shoot biomass under gradual soil drying. Generally, crops respond to water deficit by reducing water loss and maintaining turgor by stomatal closure [28]. Nevertheless, our results in experiment 2 indicated that the effect of gradual soil drying on the reduction of *gs* was primarily caused by low temperature for all crops, and their corresponding shoot biomass except for rice. Stomatal closure under drought and cold stress conditions was affected by water stress as a hydraulic activity in roots decreases [28,42]. Exp. 1A had a considerate higher temperature than Exp. 1B; however, the impact of gradual soil drying on shoot biomass of maize, sorghum, and rice in this study could not be explained by temperature as the results showed in experiment 2 (Figure 4a). The *gs* of all crops under combination of gradual soil drying and low temperature was significantly reduced than in high temperature, and rice showed a positive response as its shoot biomass was promoted by *A* (Figure 4c). In these conditions, the alternative response between *gs* and *A* of rice (C_3_) under the combination of gradual soil drying and low temperature suggested that their correlation is sometimes not positive. Furthermore, rice, a C_3_ crop, had a lower optimum temperature, and it had better CO_2_ assimilation than C_4_ crops such as maize, sorghum, and millet [62]. Cold-adapted plants displayed an increase in *A* below the optimum thermal temperature and a reduction in *A* above the thermal optimum [62,63,64,65]. In maize, sorghum, and millet, a combination of gradual soil drying and high temperature was highly promoted the shoot biomass, *A*, and *gs* (Figure 4a,c,e), but shoot biomass, *A*, and *gs* of rice decreased under a combination of gradual soil drying and high temperature. Day by day, the stomata react to changing water and temperature variables [51]; therefore, managing the responsiveness of *gs* offers breeders the potential to manage the interaction *gs* and *A*, which would impact yield [66].

### 3.2. A Plant’s Ability to Maintain Gas Exchange Is Important for Maintaining the Biomass Production

*gs*, *A*, and *E* under water and temperature variability for all crops were significantly correlated (*p* < 0.001) (Figure 6), but in rice, the coefficient correlation between *gs* and *A* was low (Figure 6d). Reactive *gs* and *A* of rice (C_3_) was indeed different from maize, sorghum, and millet (C_4_), measured at the same environmental factor [62,67]. The changing of the gas exchange clarified the effect of soil moisture status and temperature viability in experiment 2 (Figure 5). A reduction was caused by declining *gs* to prevent desiccation [68,69,70]. Under water deficit, the leaf gradually increases water potential with depletion of soil moisture content [71]. Plants increase ABA hormone concentration in their leaf, which governs close *gs* and inhibition *A* [72]. Alternately, leaf water potential is not remarkedly different under soil waterlogging [68]. It relates to limiting root respiration due to hypoxia and reducing *gs* at the early growth stage compared to water deficit [73,74].

Similarly, *gs* of maize and sorghum under combination of waterlogging and low or high temperature was declined earlier after imposed soil waterlogging compared to combination of moderated soil moisture and high temperature and gradual soil drying and high temperature. Alternatively, the *gs* of millet under combination of waterlogging and low temperature, and gradual soil drying and low temperature were also reduced earlier than the higher temperature at the same soil moisture states. This evidence suggested that the delay of *gs* leads to maintained *A* and consequently shoot biomass under water stress and temperature variability. In contrast, multi-water stress and low temperature had a higher impact on reducing *gs*, *A*, and consequently shoot biomass of maize, sorghum, and millet compared to the combination of water stresses and at higher temperature. Therefore, to consider how crops cope with the water and temperature variability of current global climate change, the ability to maintain *gs* should be a crucial parameter.

### 3.3. The Influence of Soil Moisture Content, Air Temperature and gs on Shoot Biomass of Each Crop

According to multiple linear regression, sorghum was the highest adjusted R^2^, followed by maize and millet, whereas rice was considerately lowest (Table 2). The developed crop growth models have been variable, but their effectiveness is only a specific environment and crop, and excludes the gas exchange parameter [75]. Global climate change and water and temperature stress events are predicted to increase with greater frequency or duration [40]. Thus, our crop growth model is useful for estimating multiple crops such as sorghum, maize, and millet, but not rice, under a wide range of soil water statuses and atmospheric environments. This model may therefore be considered for application in further research and irrigation schedules.

## 4. Materials and Methods

### 4.1. Seedling Preparation

Four crop species: (1) maize (*Zea mays* L. cv. Honey Bantam), (2) sorghum (*Sorghum bicolor* Moench. cv. High grain sorghum; prone to waterlogged soil but adaptable to dry soil), (3) millet (*Echinochloa utilis* Ohwi. cv. Kumamoto local), and (4) rice (interspecific progeny cv. NERICA1), as adaptable to saturated and dry soil conditions [19], were used. Each crop’s seed was placed in a Petri dish containing filter paper moistened with distilled water and left to germinate at 28 °C in an incubator under dark conditions for 2–3 days. Then, the germinated seeds were sown in a seedling tray (59 × 30 cm, containing 128 holes) filled with soil and vermiculite mixture (3:1/*v*:*v*). Ten-day-old seedlings of each crop were transplanted into experimental sites.

### 4.2. Experiment 1: Four Crops on Nine Different Water Conditions

This experiment was conducted at screen house (without atmospheric environment controlling), Kagoshima University (31.5699° N, 130.5443° E), Japan, and repeated twice (Exp. 1A and 1B). Exp. 1A and 1B were carried out in early to mid-summer (25 August–9 September 2020) and late summer to early autumn (16 October–11 November 2020).

#### 4.2.1. Experimental Site

The seedlings were grown on a concrete container (360–cm L × 110 cm W × 35–91 cm D) filled with a mixture of loamy soil and river sand (1:3 *v*/*v*). The container was divided into nine plots representative of different top sequence positions. The lowest to highest top sequence positions ranged from 30 to 90 cm, and the difference between each plot (top sequence position) was 6.5 cm.

#### 4.2.2. Treatment

Each plot was divided into three replications measuring 36.6 × 41.0 cm. Two seedlings per crop were randomly transplanted into each replication with plant interval and between row spacing at 10.0 × 13.3 cm. Rice plants were transplanted as a guard row along the borders. Daily irrigation was applied in the morning and evening to allow adequate soil moisture prior to initiate treatments.

The water treatments started the early growth stage 10 days after transplanting; leaf age was 2.5 leaves for rice and 3 leaves for other crops. The treatment was ended 17 days after treatment (DAT). Water was added to the container, allowing the lowest end to be flooded and water level maintained at 2–3 cm above the soil surface. Another soil surface of eight treatments was close to or above the water level [75]. Nine water treatment regimes were controlled in each treatment, categorized into three soil moisture statuses: dry, moderate, and wet. Three positions (sub-soil moisture status), namely, high, middle, and low, were contained in each soil moisture status. Details of the treatment are shown in Table 3.

#### 4.2.3. Soil moisture Content, Leaf Area, Shoot Biomass, and *gs*

A soil moisture sensor (5TE) placed at a depth of 15 cm was used to measure the soil moisture status of each plot (total nine plots). Data were recorded using a Datalogger Em50 Series (Decagon Devices Inc., Pullman, WA, USA) with a 60 min interval between each measurement through the experiment. Using a porometer (AP4, Delta-T Devices, Cambridge, UK) between 9:00 a.m. and 12:00 p.m. at 16 DAT, the *gs* was measured from the second youngest fully expanded leaf. The sampled shoot biomass and LA were conducted at 17 DAT by cutting the shoot and separating the leaves and stems. Then, the gathered leaves and stems were oven-dried at 80 °C to a constant weight before determining shoot dry weight. An automatic area meter (AAM-9, Hayashi Denko Co., Ltd., Tokyo, Japan) was used to measure LA.

### 4.3. Experiment 2: The Effect of Soil Water Statuses and Temperature Combination on Four Crops

This experiment confirmed the crop response to a combination of water stress and temperatures, referred to as experiment 1. This experiment was conducted at Kagoshima University, Japan, in December 2021.

#### 4.3.1. Experimental Site and Growing Media Preparation

The plants were grown with maximum photosynthetic photon in controlled environment chambers (Biotron NK system, model LPH-411PFQDT-SP; Nippon Medical and Chemical Instruments Co., Ltd., Osaka, Japan) with a flux density (MPPFD) of 930 μmolm^−2^s^−1^. The air temperature was set to 32/22 °C (day/night) with a relative humidity of 50/80% (day/night) and a light/dark regime of 12/12 h before treatment; a pot (42 cm × 28 cm × 21 cm) was filled with mixed soil containing 30% (*v*/*v*) soil, 30% vermiculite, and 10% peat moss until 2/3 (7-kg pot^−1^). After compound fertilizer with concentration of 1.3 g of each N-P-K (8-8-8; N-P-K) per pot was mixed with the soil, the soil pH was measured with an average of 5.65. Then, the container was watered abundantly for three hours before excess water was drained overnight to obtain the soil field capacity. After the soil field capacity of soil was set, each container was weighed to obtain the initial weight. The measurement of soil moisture was the same method as experiment 1.

#### 4.3.2. Method and Treatment

The experimental treatments consisted of six combinations of soil moisture and temperature, i.e., (1) combination of moderate soil moisture and low temperature (moderate soil moisture (MSM)/24/15 °C); (2) combination of moderate soil moisture and high temperature (MSM/34/25 °C); (3) combination of gradual soil drying and low temperature (gradual soil drying (GSD)/24/15 °C); (4) combination of gradual soil drying and high temperature (GSD/34/25 °C); (5) combination of continuous soil waterlogging and low temperature (continuous soil waterlogging (CSW)/24/15 °C); and (6) combination of continuous soil waterlogging and high temperature (CSW/34/25 °C). Each treatment was replicated four times. Two seedlings (each representative replication) per pot were randomly transplanted with plant interval and between row spacing at 10.0 × 13.3 cm.

To maintain adequate soil moisture content before treatments, the watering was irrigated every evening, and the amount of daily watering was estimated by water loss on the day of watering. The containers were weighed from 4:00 to 5:00 p.m. every evening to calculate water loss under MSM and GSD. Under MSM conditions, the pot was refilled by water to compensate for the water loss and maintain the soil field capacity. Under GSD conditions, a maximum of 200 g of water loss per day was fixed; if the water loss over 200 g per day was filled with an equal amount of water lost, the soil was gradually dried for low- and high-temperature treatments. Lastly, the flooded water level was set at 2–3 cm above the soil surface for CSW. The treatment ended at 17 DAT.

#### 4.3.3. Shoot Biomass, LA, *A*, *gs*, and *E*

Three plants from each treatment at 17 DAT were selected from each growth chamber to determine the *A*, *gs*, and *E*. Using a portable gas exchange measurement system (LI-6400, Li-Cor Inc., Lincoln, NE, USA) equipped with the standard leaf chamber (chamber area of 6 cm^2^), gas exchange parameters were measured on the attached second youngest fully expanded leaf at 0, 4, 8, 12, and 17 DAT from 10:00 a.m. to 2:00 p.m. The measurement settings included a light intensity of 830 mol m^−2^ s^−2^, an ambient CO_2_ concentration of 420 mol mol^−1^, and a block temperature of 27 °C for 0 days of all treatments: 19 °C for treatment of any soil moisture status under low temp treatments, and 29 °C for treatment of any soil moisture status under high temperature. The humidity was set to alter close to the growth chamber. *WUE* was calculated as ratio between *A* and *E*. The LA and shoot biomass measurement was conducted with the same procedure as experiment 1.

### 4.4. Data Analysis

All parameters were transformed using standardization to compare the shoot biomass, LA, *A*, *gs*, *E*, and *WUE* between the crops. Then, a two-way analysis of variance (ANOVA) was used for both experiment 1 and 2 to compare the crop response to treatments using Graph Pad Prism 9.0 (GraphPad Software, San Diego, CA, USA; https://www.graphpad.com (accessed on 11 March 2022)). The linear or non-linear (polynomial) correlation line was used, which was decided by coefficient. Pearson’s correlation was conducted to test the significant correlation of linear or non-linear correlation. A multiple linear regression was used with single and combination parameters among soil moisture status, temperature, *A*, *gs*, *E*, and *WUE* to evaluate which factors influenced shoot biomass. It can be used for multiple crops. Turkey’s test was used to test the statistical differences among the treatments.

## 5. Conclusions

Different crops responded differently to different soil moisture, temperature, and these two stresses in combination. Decreased stomatal conductance and biomass accumulation was observed, and the highest decrease was observed when crops were exposed to combined stress. However, the effect of these stresses varied among the crop genotypes. The combination of various soil water status and temperature variation, rice, and maize were less effective on biomass production compared to millet and sorghum. Biomass accumulation of all crop genotypes was reduced by all treatments compared to optimal growing condition (i.e., moderate temperature in the presence of adequate temperature). Maize and sorghum under waterlogging conditions reduced shoot biomass, presumably due to the decreased stomatal conductance and photosynthesis, which was temperature independent, whereas for rice and millet, the reduction was also due to decreased stomatal conductance; it was temperature dependent. All crops indicated temperature-dependent stomatal conductance (at GSD/34/25 °C), where the *gs* of rice was lowest under high temperature. Thus, our results suggest that an ability to sustain *gs* is essential for photo assimilation and maintaining leaf temperature through evapotranspiration for biomass production, a mechanism of crop avoidance to combine variable soil water status and temperature.

## Figures and Tables

**Figure 1 plants-11-01039-f001:**
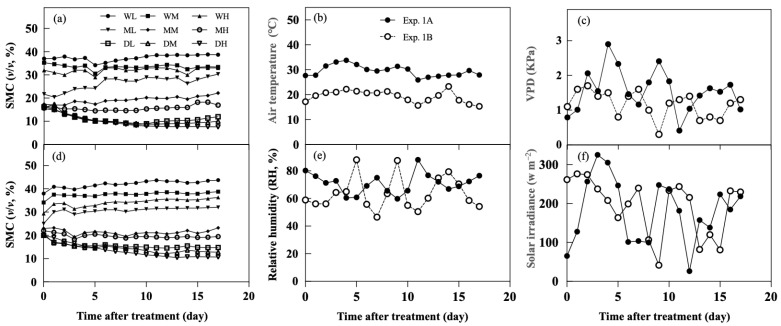
Soil moisture content (SMC) in experiment 1A (**a**) and 1B (**d**), air temperature (**b**), vapor pressure deficit (VPD) (**c**), relative humidity (**e**), solar irradiance (**f**), and during the treatment of experiments 1A and 1B.

**Figure 2 plants-11-01039-f002:**
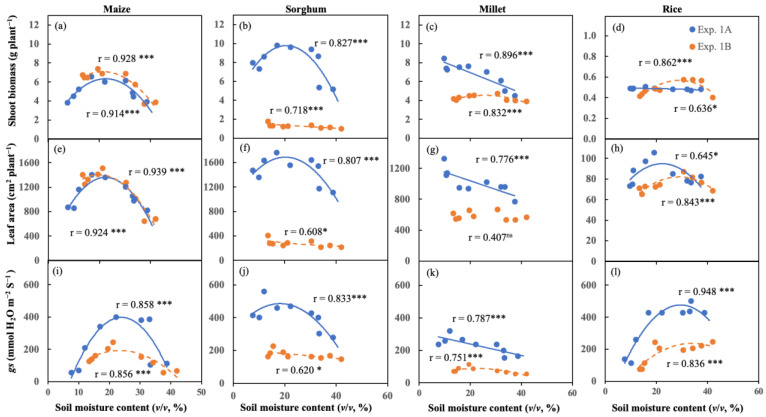
Correlations between soil moisture content and shoot biomass (**a**–**d**), leaf area (**e**–**h**), and stomatal conductance (*gs*; **i**–**l**) in maize (**a**,**e**,**i**), sorghum (**b**,**f**,**j**), millet (**c**,**g**,**k**), and rice (**d**,**h**,**l**). *, ***, and ns indicate Pearson statistical significance at *p* < 0.05, *p* < 0.001, and non-significance, respectively (*n* = 9). Linear or nonlinear (polynomial) correlation line is decided by coefficient.

**Figure 3 plants-11-01039-f003:**
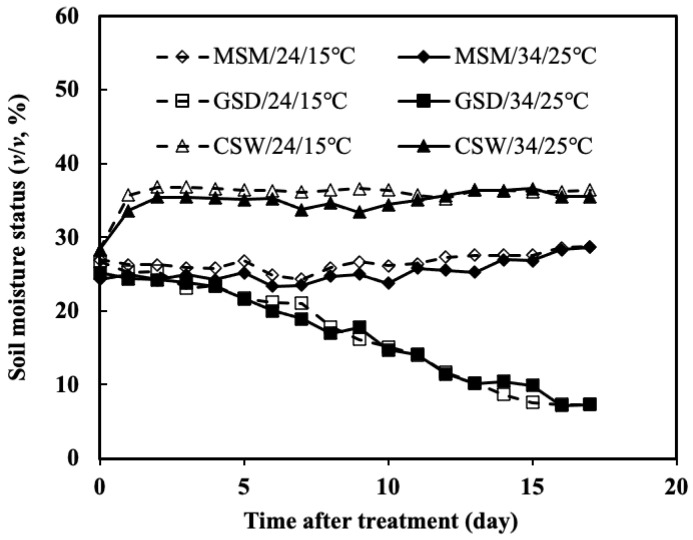
Soil moisture content (SMC) during the treatment period of experiment 2.

**Figure 4 plants-11-01039-f004:**
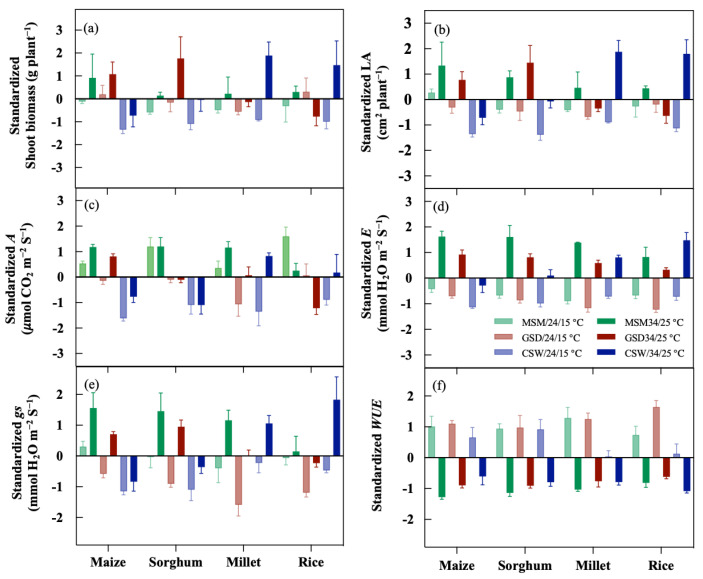
The responses of shoot biomass (**a**), leaf area (LA; (**b**)), photosynthesis rate (*A*; (**c**)), transpiration rate (*E*; (**d**)), stomatal conductance (*gs*; (**e**)), and water use efficiency (*WUE*; (**f**)) of maize, sorghum, millet, and rice to the combination of soil moisture contents and temperatures by standardization data. The shoot biomass and LA were at 17 days after treatment, while gas exchange (*A*, *gs*, *E*, and *WUE*) was the average of all measurements after treatment. Bars indicate mean standard deviation. Standardization was used for transformation of data.

**Figure 5 plants-11-01039-f005:**
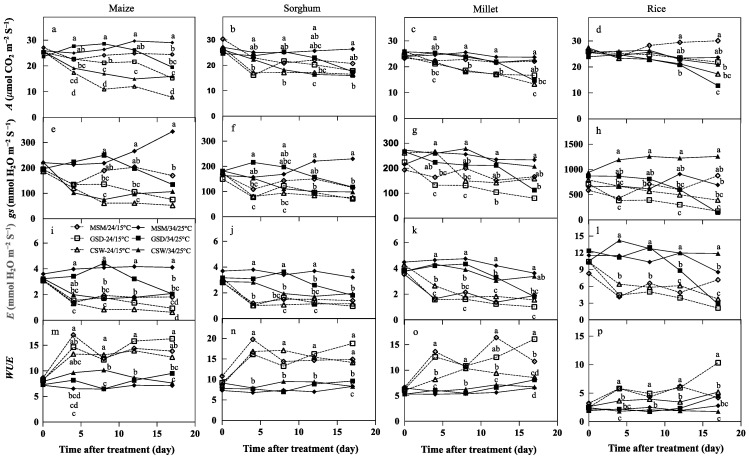
The effect of combination of soil moisture status and temperature on changing of photosynthesis rate (*A*; (**a**–**d**)), stomatal conductance (*gs*; (**e**–**h**)), transpiration rate (*E*; (**i**–**l**)), and water use efficiency (*WUE*; (**m**–**p**)) in maize (**a**,**e**,**i**,**m**), sorghum (**b**,**f**,**j**,**n**), common millet (**c**,**g**,**k**,**o**), and rice (**d**,**h**,**l**,**p**) during the course of experiment 2. Each day of measurement with similar letters did not significantly differ according to Tukey’s test at the 0.05 probability level.

**Figure 6 plants-11-01039-f006:**
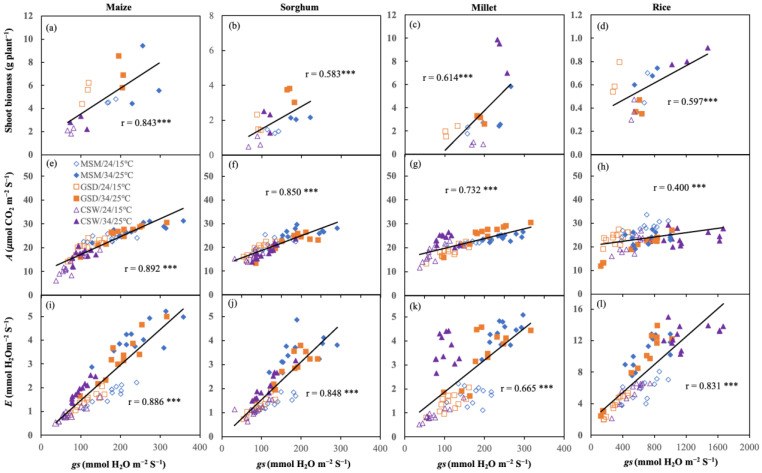
Correlation between stomatal conductance (*gs*), shoot biomass (**a**–**d**), and photosynthesis rate (*A*; (**e**–**h**)), and transpiration rate (*E*; (**i**–**l**)) in maize (**a**,**e**,**i**), sorghum (**b**,**f**,**j**), millet (**c**,**g**,**k**)), and rice (**d**,**h**,**l**). *** indicates Pearson statistical significance at *p* < 0.001 (*n* = 18 for shoot biomass and *n* = 72 for *A* and *E*).

**Table 1 plants-11-01039-t001:** The effect of the combination of environmental factors (soil moisture status and temperature) on shoot biomass, LA, *A*, *gs*, *E*, and *WUE* of crops (maize, sorghum, millet, and rice), and interaction between crops and environment in experiment 2.

Source of Variation	Shoot Biomass	Leaf Area	*A*	*gs*	*E*	*WUE*
Crops	ns	ns	ns	ns	ns	ns
Soil water status (SWS)	***	***	***	***	***	***
Crops × soil water status	***	***	***	***	***	***

*** and ns indicate statistical significance of ANOVA at *p* < 0.001 and non-significance, respectively.

**Table 2 plants-11-01039-t002:** The multiple linear regression for shoot biomass (g plant^−1^) based on parameters of temperature (Temp), soil moisture content (SMC), and stomatal conductance (*gs*) under three soil moisture regimes (MSM, GSD, CSW) and two temperatures (24/15 °C and 34/25 °C) (*n* = 18).

Maize						Sorghum					Millet					Rice				
Equation	Variation	β	*t*-Value	Adjusted R^2^	*p*-Value	Equation	β	*t*-value	Adjusted R^2^	*p*-Value	Equation	β	*t*-Value	Adjusted R^2^	*p*-Value	Equation	β	*t*-Value	Adjusted R^2^	*p*-Value
(1)	Intercept	5.71	2.906	0.658	0.000	(2)	0.947	1.351	0.759	0.000	(3)	−9.832	−3.072	0.492	0.006	(4)	0.556	1.794	0.262	0.066
	Temp	0.071	0.983				0.163	4.875				0.455	2.514				−0.006	−0.567		
	SMC	−0.171	−3.271				−0.085	−4.602				0.208	2.126				−0.008	−0.896		
	*gs*	0.012	1.877				−0.005	−1.293				−0.016	−0.750				0.001	2.231		

**Table 3 plants-11-01039-t003:** Explanation of soil moisture statuses (treatments) in experiment 1.

No.	Abbreviation	Soil Water Statuses (Treatments)
1	WL	Low position of sub-wet soil conditions (waterlogging)
2	WM	Middle position of sub-wet soil conditions
3	WH	High position of sub-wet soil conditions
4	ML	Low position of sub-moderate soil conditions
5	MM	Middle position of sub-moderate soil conditions
6	MH	High position of sub-moderate soil conditions
7	DL	Low position of sub-dry soil conditions
8	DM	Middle position of sub-dry soil conditions
9	DH	High position of sub-dry soil conditions (severe dry soil)

## Data Availability

The data presented in this study are available on request from the corresponding author.

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
