# Peer review of "Combinational Variation Temperature and Soil Water Response of Stomata and Biomass Production in Maize, Millet, Sorghum and Rice"

_plants, 2022, doi:10.3390/plants11081039_

Round 1
Reviewer 1 Report
The authors present a study that highlights how different important crops respond to combinations of soil moisture and environmental conditions through the use of stomatal conductance and carbon assimilation as a proxy to photosynthetic activity.
The paper is thorough and at times a little bit tedious. I would recommend for the authors to think if all the figures are needed in order to convey their message to the reader.
also, the English could benefit from additional editing.
Author Response
Authors respond to a review:
1). Title, abstract, introduction, result, discussion, and conclusion were revised.
2). The English language had edited.
5). The figure 2, 8, and 9 were removed due to overlap information and not suitable.
6). Figures were arranged in order.

Reviewer 2 Report
This manuscript entitles “Combinational Variation Temperature and Soil Water Response of Stomatal Control of Transpiration in Miaze, Common Millet, Sorghum, and Rice” provides some information on the interaction of temperature and soil water content on the stomatal conductance of four cereal crops. This manuscript has a lot of mistakes and required extensive English editing for readability.
Comments
- Title “Combinational Variation Temperature and Soil Water Response of Stomatal Control of Transpiration in Miaze, Common Millet, Sorghum, and Rice”. Spelling mistake, it should be “maize”.
- “… and parched soil (DH) “ revise to “dry soil” same with L91.
- L96, “… but the their fluctuations were observed as differencing between Exp.1A and 1B (Figure 1c, e and f). Please revise this sentence.
- L101, “Figure 2 demonstrates a positive or negative number that means the shoot biomss was higher or lower than mean of all soil moisture treatments, respectively” please revise to “Figure 2 showed a positive or negative “values” indicated that the shoot “biomass” was higher or lower than mean of all soil moisture treatments, respectively ”
- L113, revise “but it was not so for rice under the same conditions.”
- L147, should be “polynomial”.
- L199, Fighure 5f explaines that low temperature promoted a positive reponse of WUE under numberous soil moisture status, but high temperature negatively impacted WUE in all crop. Please revise to “Figure 5f showed that…response of WUE under different soil…. All crops”
- L203, The change in gas exchange during the treatment period of the individual crop across the combination between soil moisture status and temperature treatments is shown in 204 Figure 6. Please revise this sentence. Not clear.
- L284, revise Figure 6. The effect of soil moisture status and temperature on the changing of photosynthesis rate (A; a⎯d), …
- L296, “itemize”? Please revise.
- L298, revise sentence “Plant mechanism of response to water stress includes conservative and non-conservative [43]. ”. What is the non-conservative mechanism?
- L305-307, revise this sentence.
- L397, “And it considers to apply for further research, irrigation schedules, or technology transfer. ” Why technology transfer here?
- Please adjust Table 2.
- L498, “A/E calculated WUE.” Should be revised to “WUE was calculated as the ratio between A and E”. Also need to revise other places in the text.
- Please revise the conclusion section.
Author Response
Authors respond to a review:
1). Title, abstract, introduction, result, discussion and conclusion were revised.
2). The English language had edited.
5). The figure 2, 8, and 9 were removed due to overlap information and not suitable.
6). Figures were arranged in order.
Comments
1. Title “Combinational Variation Temperature and Soil Water Response of Stomatal Control of Transpiration in Miaze, Common Millet, Sorghum, and Rice”. Spelling mistake, it should be “maize”.
Response1: agree “maize”
2. “… and parched soil (DH) “ revise to “dry soil” same with L91.
Response2: agree “parched soil (DH) revised to dry soil (DH)” (L145).
3. L96, “… but the their fluctuations were observed as differencing between Exp.1A and 1B (Figure 1c, e and f). Please revise this sentence.
Response3: Yes, already revised (152-154).
4. L101, “Figure 2 demonstrates a positive or negative number that means the shoot biomss was higher or lower than mean of all soil moisture treatments, respectively” please revise to “Figure 2 showed a positive or negative “values” indicated that the shoot “biomass” was higher or lower than mean of all soil moisture treatments, respectively ”
Response 4: The figure 2 was removed, and this part had also removed.
5. L113, revise “but it was not so for rice under the same conditions.”
Response 5: Yes.
6. L147, should be “polynomial”.
Response 6: Yes, agree. It should be “polynomial” (L204)
7. L199, Fighure 5f explaines that low temperature promoted a positive reponse of WUE under numberous soil moisture status, but high temperature negatively impacted WUE in all crop. Please revise to “Figure 5f showed that…response of WUE under different soil…. All crops”
Response 7: Yes, agree. It already revised to “Figure 4f showed that low temperature promoted a positive response of WUE under numerous soil moisture status, but high temperature negatively impacted WUE in all crop”. (L255-257)
8. L203, The change in gas exchange during the treatment period of the individual crop across the combination between soil moisture status and temperature treatments is shown in 204 Figure 6. Please revise this sentence. Not clear.
Response 8: Yes, already revised. Noted “Figure 6 changed to Figure 5” (L259-261).
9. L284, revise Figure 6. The effect of soil moisture status and temperature on the changing of photosynthesis rate (A; a⎯d), …
Response 9: Figure 6. The effect of combination of soil moisture status and temperature on changing of photosynthesis rate (A; a⎯d), stomatal conductance (gs; e⎯h), transpiration rate (E; i⎯l), and water use efficiency (WUE; m⎯p) in maize (a, e, i, and m), sorghum (b, f, j, and n), common millet (c, g, k, and o), and rice (d, h, l, and p) during the course of experiment 2. Each day of measurement with similar letters did not significantly differ according to Tukey’s test at the 0.05 probability level. (L329-332)
10. L296, “itemize”? Please revise.
Response 10: itemize revised to “optimize” (L359)
11. L298, revise sentence “Plant mechanism of response to water stress includes conservative and non-conservative [43]. ”. What is the non-conservative mechanism?
Response 11: Non-conservation mechanism is closely referred to slower opening and closing stomata under drought, while the conservative mechanism is closely referred to faster closing stomata under drought. (L362-364)
12. L305-307, revise this sentence.
Response 12: this sentence was removed due to remove the figure 2
13. L397, “And it considers to apply for further research, irrigation schedules, or technology transfer. ” Why technology transfer here?
Response 13: “or technology transfer” was removed. And the sentence is modified to show how the model can be applied at farm level for water management and estimation the growth of crops. (L466-467)
14. Please adjust Table 2.
Response 14: Yes, it was adjusted.
15. L498, “A/E calculated WUE.” Should be revised to “WUE was calculated as the ratio between A and E”. Also need to revise other places in the text.
Response 15: Yes, it already revised. (L568)
16. Please revise the conclusion section.
Response 16: Yes, it already revised.

Round 2
Reviewer 2 Report
Thanks for the great improvement. However, the title mentioned "Stomata and Biomass Production...". It will be nice if Fig.6 can include a statistical analysis of biomass data (Fig.1) with gas exchange data (gs etc.) to make a solid scientific conclusion.
Author Response
Comments and Suggestions for Authors
Thanks for the great improvement. However, the title mentioned "Stomata and Biomass Production...". It will be nice if Fig.6 can include a statistical analysis of biomass data (Fig.1) with gas exchange data (gs etc.) to make a solid scientific conclusion.
Authors respond to a reviewer: Thank you very much for good comments and contribution to our manuscript.
- The correlation between gs and shoot biomass has included in Figure 6a-d.
- The explanation of figure6 has modified (L317-324).
